# Cast Structure in Alloy A286, an Iron-Nickel Based Superalloy

**Robin Frisk \*, Nils Å. I. Andersson  and Bo Rogberg**

Department of Materials Science and Engineering, KTH Royal Institute of Technology, SE-100 44 Stockholm, Sweden; nilsande@kth.se (N.Å.I.A.); brogberg@kth.se (B.R.)
* Correspondence: rfrisk@kth.se

**Abstract:** The structure and segregation of a continuously cast iron-nickel based superalloy were investigated. Cross-sectional samples were prepared from the central section of a $150 \times 150$ mm square billet. The microporosity was measured from the surface to the center and theoretical conditions for pore formation were investigated. A central porosity, up to 10 mm in width, was present in the center of the billet. The measured secondary arm spacing was correlated with a calculated cooling rate and a mathematical model was obtained. Spinel particles were found in the structure, which acted as inoculation points for primary austenite and promoted the formation of the central equiaxed zone. Titanium segregated severely in the interdendritic areas and an increase of Ti most likely lead to a significant decrease in the hot ductility. Precipitates were detected in an area fraction of approximately 0.55% across the billet, which were identified as Ti(CN), TiN, η-Ni3Ti, and a phosphide phase.

**Keywords:** A286 superalloy; cast structure; continuous casting; porosity; intermetallic phases; dendrite arm spacing

## 1. Introduction

The austenitic stainless steels are the most commonly produced group of stainless steels. The stability of the mechanical properties of austenitic stainless steels makes them suitable for high-temperature applications, as well as for cryogenic applications [1]. Due to their face-centered cubic crystal structure, they are ductile even at extremely low temperatures. Furthermore, they have a good corrosion resistance due to their high chromium content.

The A286 steel alloy is an iron-nickel based austenitic superalloy with a Ti content of 2 wt % to promote a uniform formation of strengthening precipitates [2]. The finished products are strengthened mainly by precipitation of $\gamma'$-Ni$_3$(Al,Ti) [3] after a heat treatment carried out at temperatures between 615 to 825 °C [4,5]. However, the mechanical properties degrade if the alloy is treated above 750 °C for a prolonged time, due to the formation of an η-Ni$_3$Ti phase [4–6]. The alloy has good corrosion resistance and creep strength at elevated temperatures and it is used in various high-temperature applications such as in gas turbines, jet engines, and turbo chargers as blades, frames, casings, and fasteners [7].

Several previous authors have reported on the structure of rolled A286 and the precipitation kinetics of $\gamma'$ and η, after one or several heat treatments. Several titanium compounds such as TiC, TiN, Ti(CN), Ti$_4$C$_2$S$_2$, Laves-Fe$_2$Ti, and M$_{23}$C$_6$ are found in solution treated A286 alloys [3–6,8–11]. The solidification structure and segregation formed during casting determines how well the material is deformed in subsequent process operations such as hot working [12–14]. The presence of a Ti rich phase in the grain boundaries has been reported to cause a molten thin film during hot ductility tests [12] and an increase of phosphorous content in austenitic stainless steels has been reported to have a high impact on the hot ductility [13]. In electroslag remelting (ESR) ingots the segregation of Ti has been examined. The results showed that with increasing Ti contents there was an increase of A segregations and

hardness. Additionally, there was lower elongations in tensile tests [14]. The solidification structure in welds has also been investigated, where a high segregation of Ti in the interdendritic areas has been found [2,15].

In the current study, the as-cast microstructure of continuously cast A286 and the phases formed during the solidification process are investigated. By characterization of the precipitates and matrix structure, an assessment is made of the consequence it might have on subsequent processes such as soaking and hot rolling. From measurements and theoretical calculations, the conditions of pore formation are discussed.

## 2. Materials and Methods

The composition of the examined material is given in Table 1. The samples were taken from a continuously cast billet with a cross section of $150 \times 150$ mm$^2$. No electromagnetic stirring was applied during casting. The three-strand continuous casting machine had a radius of 12 m and the heat size was 75 tons. The billet sample for this investigation was taken at a location corresponding to half of the total casting time for a strand.

First, a central plate of about 20 mm width was cut from the billet sample. The orientation of the plate is indicated in Figure 1. The loose side corresponds to the upper side of the billet and the fix side corresponds to the lower side. The plate was then machined to the center plane, where the center porosity was revealed. This resulted in a plate with the dimensions L × W × H. $225 \times 150 \times 10$ mm. To produce micro samples, a slice was cut from the plate and then divided into cross-sectional samples designated 1–11, where sample six included the billet center. These samples were then mechanically polished using standard procedures which are common when preparing metallographic analysis samples.

The structure of the material was investigated with respect to the macro- and microstructure, dendritic arm spacing, porosity, precipitates, inclusions, and segregation of alloying elements. The investigation was made with the help of light optical microscopy (LOM) and scanning electron microscopy (SEM) equipped with an energy-dispersive X-ray spectroscopy (EDX). The area fraction of porosity and precipitates were measured with the image analysis software ImageJ.

In order to investigate the solidification behavior of the alloy, the Thermo-Calc software version 2017a TCNI8.1 [16] was used to make thermodynamic calculations.

**Table 1.** Composition of A286 in wt %.

| Component | Ni | Cr | Ti | Mn | Mo | Si | V | Al | Cu | P | C | Fe |
|---|---|---|---|---|---|---|---|---|---|---|---|---|
| **wt %** | 24.4 | 14.5 | 2.13 | 1.29 | 1.18 | 0.21 | 0.20 | 0.19 | 0.17 | 0.018 | 0.04 | bal. |

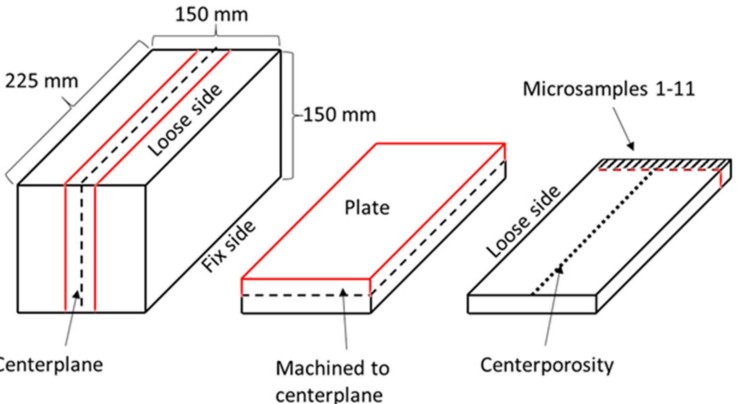

**Figure 1.** Preparation of cross-sectional samples from a 150 mm square billet sample.

## 2.1. Metallographic Studies

The porosity and the fraction of total precipitations were best determined on polished samples. To reveal the location of the precipitates and the dendritic structure of the material, all samples were swab etched for up to 5 s in Marble's reagent with the composition 4 g $CuSO_4$ + 20 mL HCl + 20 mL $H_2O$. The precipitations were then studied at 20× magnification and, for each sample, a total area of 1.4 mm$^2$ was evaluated. The LOM images were manually converted to black and white areas to separate porosities or precipitations from the matrix. Thereafter, a particle count analysis was made.

In order to make an assessment of the size and fraction of porosities in the billet from the surface to the center of the billet, 5× magnification images were taken with the aid of LOM. The investigated image area for each sample was approx. 22 mm$^2$. A sample piece of 10 × 15 mm with a thickness of 5 mm, taken from 30–45 mm below the billet surface, was dissolved using an electrical current of 70 mA in a methanol-based electrolyte for 2 h. The electrolyte was then filtrated through a polycarbonate film filter to collect the undissolved precipitates. The chemical composition of the particles was then analyzed by SEM/EDX.

The elemental mappings were made in SEM/EDX. Point analyses were made on precipitations and non-metallic inclusions. Line analyses were also made to study the microsegregation of elements across dendrite arms and an area mapping of precipitates showing the distribution of some common elements.

## 2.2. Secondary Dendrite Arm Spacing

The secondary dendrite arm spacing (SDAS) measurements were done every 5 mm transversally across the billet. Overall, secondary dendrite arms of approximately 800 dendrites were counted across the billet cross section. Thereafter, the average SDAS value was calculated using the line intercept method [17]:

$$SDAS = \frac{L}{number\ of\ arms - 1} \tag{1}$$

where, $L$ is measured distance in μm.

## 2.3. Segregation

The microsegregation of elements was studied using EDX elemental analysis. Several line analyses were made on sample 2 to the central sample 6 and an average segregation index for each sample were calculated. The segregation index, $I_i$, was defined as the measured concentration in the interdendritic areas divided by the measured concentration in the center of a dendrite:

$$I_i = \frac{C_i^{interdendritic}}{C_i^{dendritic}} \tag{2}$$

where, the value $C$ is the content in wt % of the element i: Ni, Cr, Ti, Mo, Mn.

## 3. Results

## 3.1. Structure

An image of the longitudinal section etched in Marble's reagent is presented in Figure 2. A columnar structure is seen growing as horizontal lines from the surface region. Closer to the center, the horizontal lines diminish, and thus demarks the beginning of an equiaxed structure. In addition, open porosity and distinct V-segregation pattern are seen along the center line. The center porosity consists of cavities with a width of up to 10 mm and a length of up to 30 mm. The cavities are both isolated and interconnected in the studied plane.

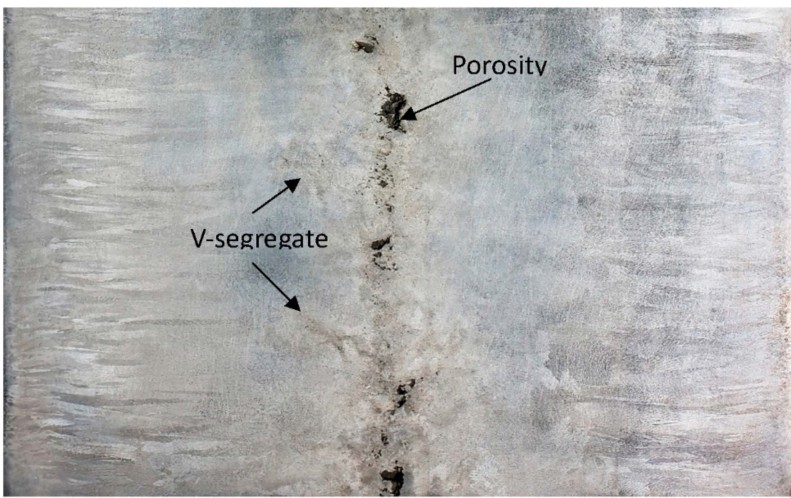

**Figure 2.** Macroetched longitudinal section of billet sample.

Images from polished samples are shown in Figure 3. The dark grey phases are precipitates of different types and the black spots are pores. Precipitates are seen to be fine and dispersed evenly, close to the surface, as seen in samples one and two. Closer to the center of the billet they become coarser. Since the precipitates are located in the interdendritic regions they are an indication of the coarseness of the dendritic structure.

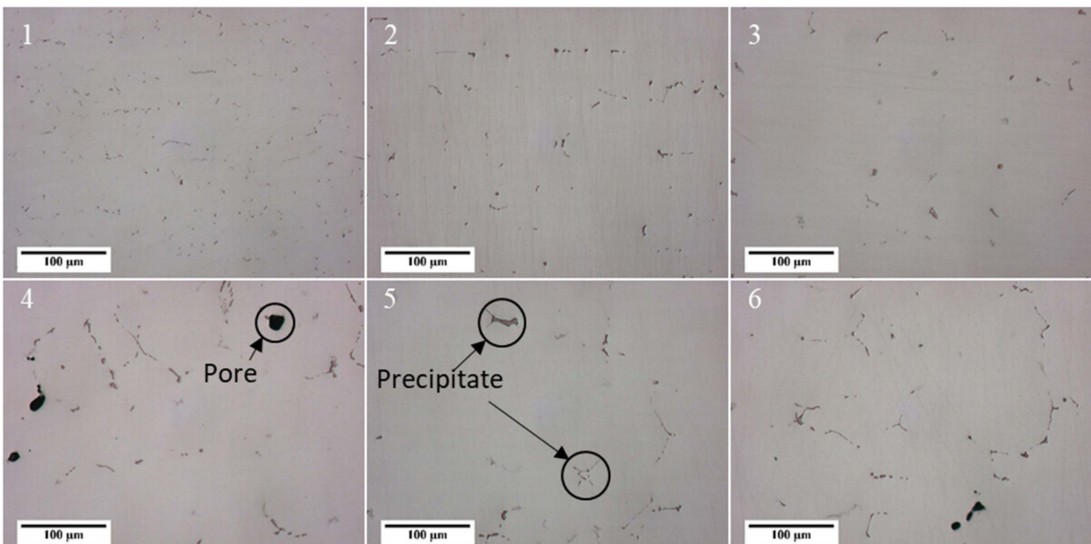

**Figure 3.** LOM images showing the precipitations as seen in polished samples from the surface to the center of the billet, at the distances; (**1**) 5 mm, (**2**) 15 mm, (**3**) 30 mm, (**4**) 45 mm, (**5**) 60 mm and (**6**) 70 mm.

Figure 4 shows the evolution of the dendrite structure on the loose side from the surface to the center of the billet. The dark areas are both precipitates and porosities which are present in the interdendritic areas and grain boundaries. However, porosities are more easily seen on the polished samples. In sample one, the change from a fine grain surface structure into a columnar structure is seen. In samples two to four, the columnar structure coarsens with an increased distance from the surface. In addition, the boundaries between columnar grains are seen to contain some precipitates. The equiaxed zone is observed to begin at approximately 50 mm from the surface at the loose side, between sample four and five in Figure 4, and at approximately 35 mm from the fix side. In the center

sample six, single dendrite arms surrounded by a continuous network of interdendritic porosity is seen adjacent to the large central pore.

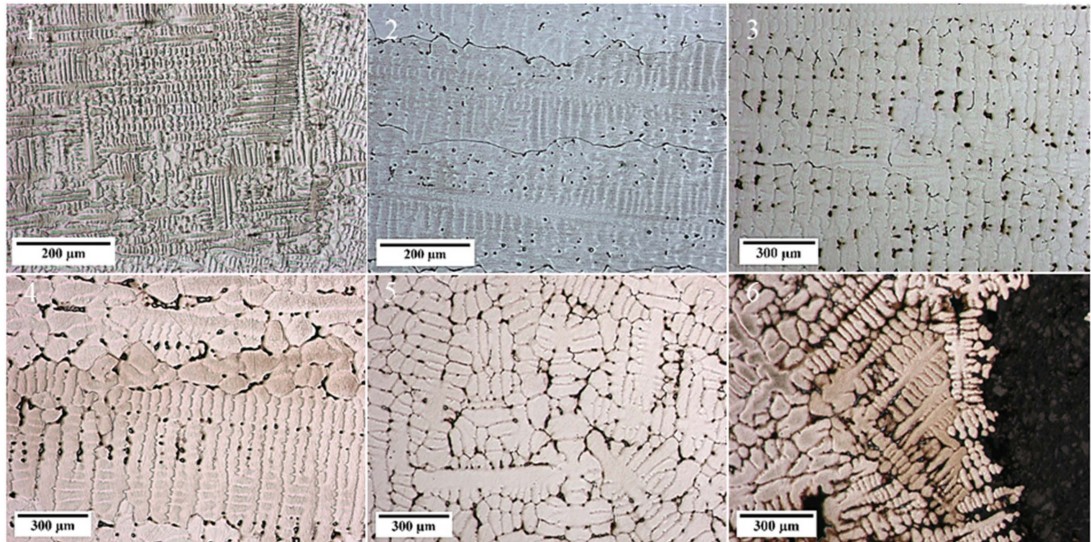

**Figure 4.** LOM images showing dendrite structure in etched samples from the surface to the center of the billet, at the distances; (**1**) 5 mm, (**2**) 15 mm, (**3**) 30 mm, (**4**) 45 mm, (**5**) 60 mm and (**6**) 70 mm.

In Figure 5, the dendrite structure at the billet surface is shown at a higher magnification. In image a, the fine structure, characteristic for the chill zone, is seen. Then, the crystals gradually increase in size further in from the surface and start to develop into the typical columnar structure. The coarser equiaxed grain, which distinguished by an arrow in Figure 5a, is captured by the solidification front. In Figure 5b, a coarser dendrite structure is seen to grow from an indentation, probably corresponding to an oscillation mark.

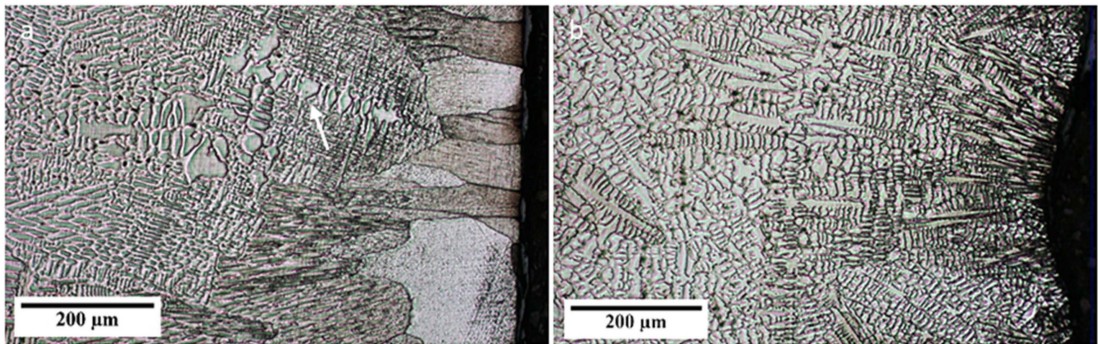

**Figure 5.** Structure of the surface region in sample 11, showing (**a**) a coarse grain next to a fine surface structure and (**b**) a coarse surface structure.

*3.2. Dendrite Arm Spacing*

The average of the measured SDAS and the standard deviation in relation to the distance from the billet surface are shown in Figure 6. The columnar and equiaxed zones are marked in the figure. The SDAS value is very small at the surface of the billet, but then it increases steadily until it reaches the equiaxed zone. In the equiaxed zone, the SDAS value decreases with an increased distance from the surface. At 30 mm from the billet surface the primary dendrite arm spacing (PDAS) was measured as 364 µm. The SDAS is proportional to the PDAS by a factor of approximately 10.

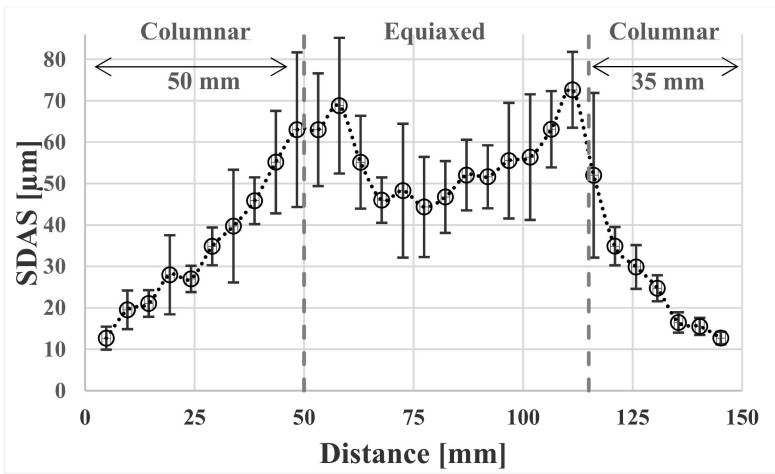

**Figure 6.** Secondary dendrite arm spacing with standard deviation measured on the cross section over the billet.

### 3.3. Porosity

Representative images of the porosity at 30, 45, 60, and 70 mm from the surface are shown in Figure 7. At approximately 30 mm from the surface (see Figure 7a) single round pores start to appear. No porosity is visible closer to the surface than 30 mm. With an increasing distance from the surface, pores become larger and more elongated. In Figure 7c the structure has changed to equiaxed. Around the center in Figure 7d, pores become larger and a connected network of interdendritic and elongated porosity is seen.

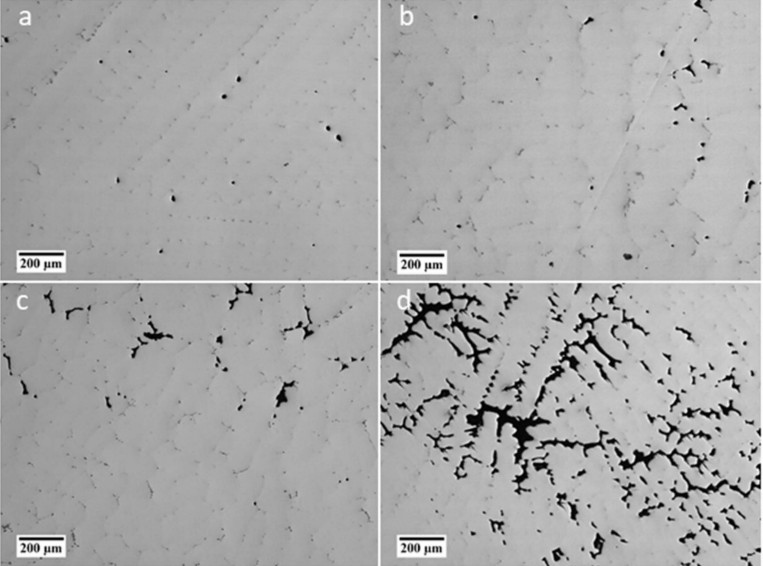

**Figure 7.** Porosity at distance: (**a**) 30 mm from surface, (**b**) 45 mm from surface, (**c**) 60 mm from surface, and (**d**) 70 mm from surface.

A 500× magnification of a pore, close to the center, with the shape of an interdendritic region is shown in Figure 8. Three equiaxed dendrites are visible around the pore. An EDX elemental analysis showed that no precipitate or unusual segregation were found adjacent to the pore boundary.

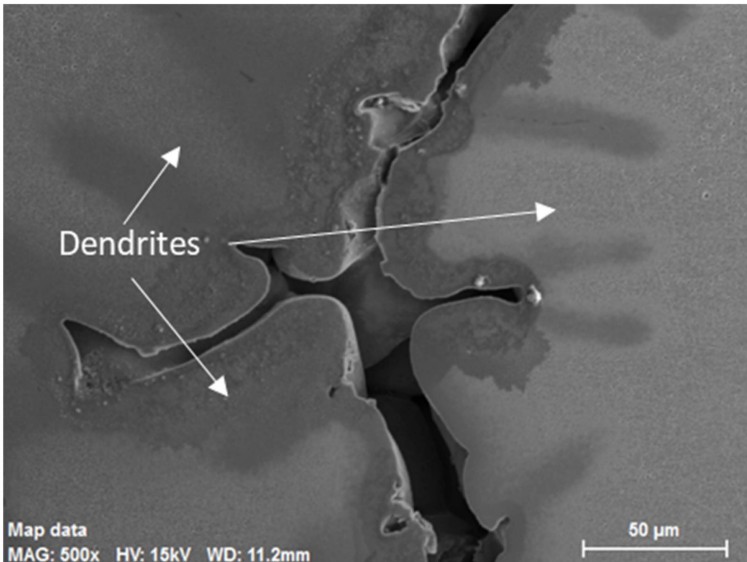

**Figure 8.** Porosity between dendrites, close to the center.

The results, from the image analysis measurements of the average radius and the fraction of pores over the cross section, are shown in Figure 9. The vertical dashed lines mark the width of the equiaxed zone. Both the area fraction and pore size increase with an increasing distance from the surface. There is a significant increase in the degree of porosity in the equiaxed zone, ~0.17%, as compared with the columnar zone, ~0.09%. The average radius value of the pores does not increase significantly upon reaching the equiaxed zone. The microporosity around the center, at 75 mm, is typically interconnected and on the verge of becoming macroporosity.

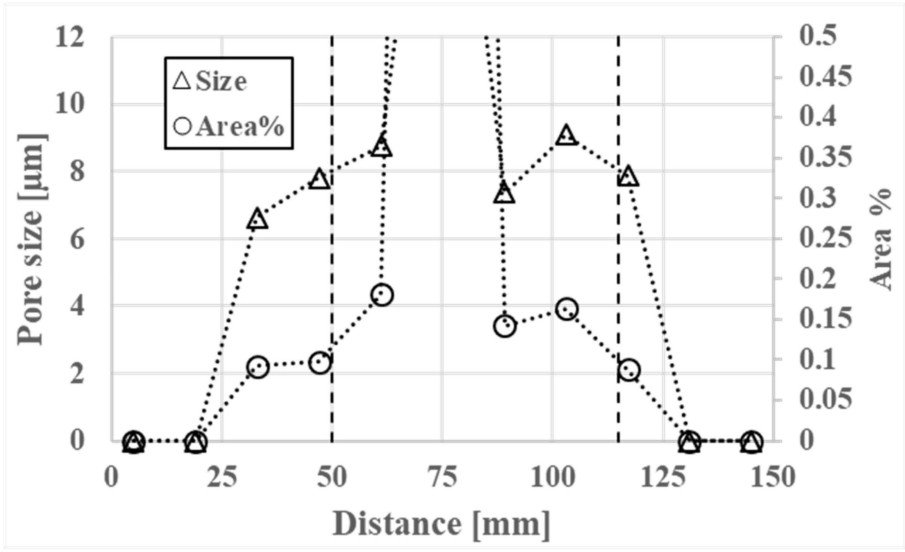

**Figure 9.** Average size and area% of analyzed porosity from the billet cross section. The horizontal lines show the width of the equiaxed zone.

### 3.4. Precipitates of Intermetallic Phases, Carbides, and Non-Metallic Inclusions

The image in Figure 10 contains the most common phases found in the material. A spot analysis was made on the designated phases and the results are presented in Table 2. The bright grey phase, denoted as one, was rich in Ni and Ti and was often found connected to other precipitates as acicular platelets. The dark grey phase, denoted as two, was rich in Ti, Fe, and P, and was found in the same places as the black phase, and larger precipitates of this type could sometimes be found in intersections

with three grain boundaries. The black phase, denoted as three, was rich in Ti and C, and was present in the interdendritic areas and in the grain boundaries. The accuracy of analyzed C content is uncertain when using EDX, however the result was used qualitatively to determine the identity of the compounds. In Figure 11 an elemental mapping of the area seen in Figure 10 shows that the titanium rich phase also contains some N and that the high C content only exist in certain precipitates.

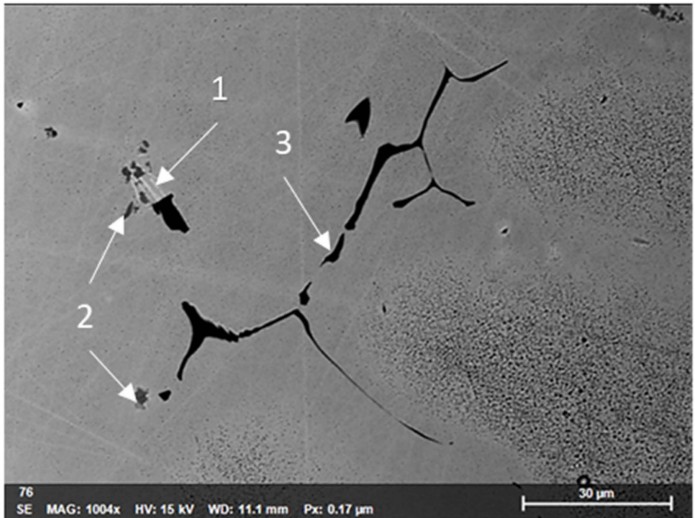

**Figure 10.** Images of common precipitates.

**Table 2.** X-ray spectroscopy (EDX) spot analysis result shown in atom%.

| Composition in at % | C | Si | P | Ti | Cr | Mn | Fe | Ni | Mo |
|---|---|---|---|---|---|---|---|---|---|
| 1-Nickel rich bright grey phase | - | 0.27 | - | 20.8 | 2.86 | 0.71 | 15.8 | 59.05 | - |
| 2-Phosphorous rich dark grey phase | - | 1.02 | 18.2 | 36.3 | 5.37 | - | 24.1 | 9.78 | 0.55 |
| 3-Titanium rich black phase | 33.7 | 0.09 | - | 58 | 1.24 | - | 2.79 | 1.61 | 2.51 |

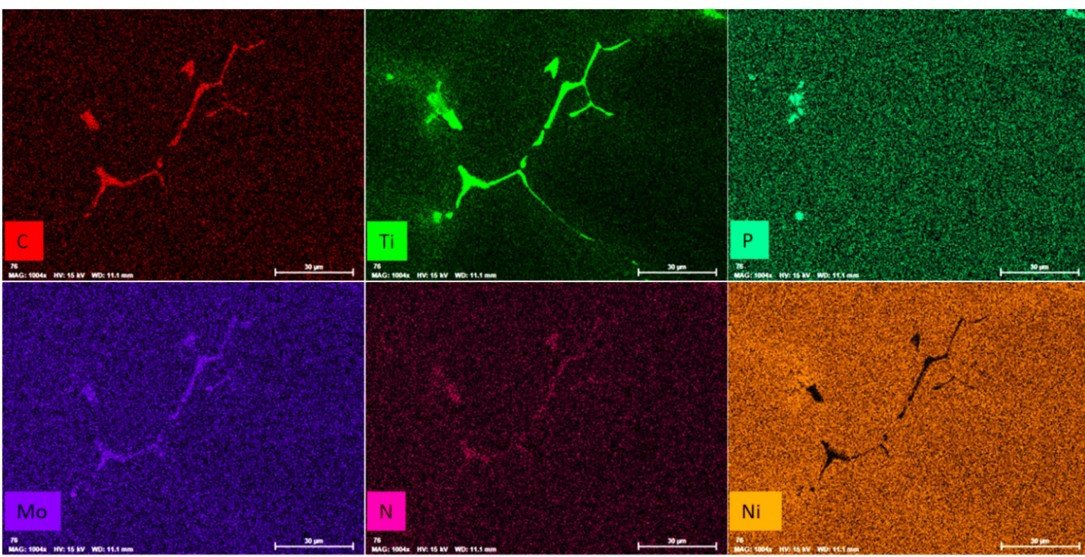

**Figure 11.** Elemental mapping of C, Ti, P, Mo, N, and Ni content in precipitates.

The EDX analysis, in Table 3, of the extracted precipitates, seen in Figure 12, gives a composition that is not affected by the surrounding matrix phase, although the carbon content of the film filter adds to the already existing uncertainty when analyzing carbon with EDX. The left image shows a thin

flake precipitate which is only composed of Ti, C, and some Mo. The right image shows an irregularly formed precipitate resembling a eutectic structure, with high P, Ti, Si, and S contents. Several analyses on other particles, such as the one in the right image, show a high variance in composition but it is always high in P and Ti.

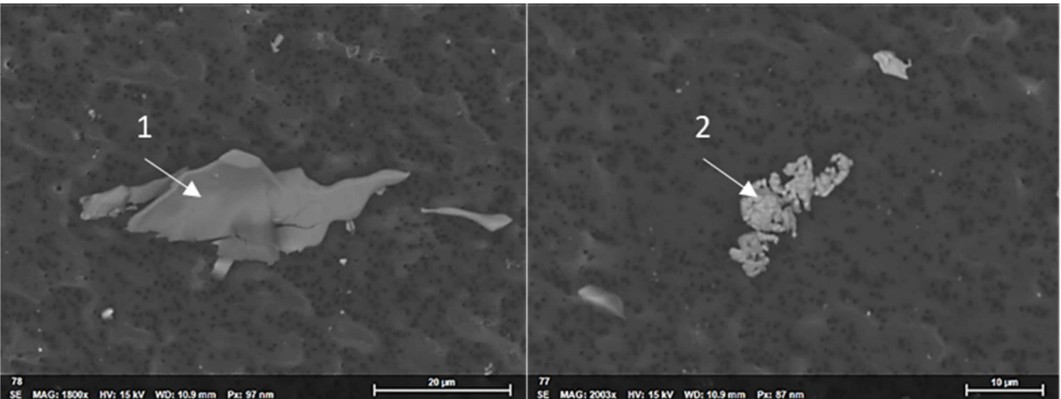

**Figure 12.** Extracted precipitates showing (**1**) the morphology of a TiC as a thin flake and (**2**) a phoshide with an irregular cellular form.

**Table 3.** EDX analysis on extracted particles.

| Composition in at % | C | Si | P | Ti | Mo | Ni | Cr | Fe | S |
|---|---|---|---|---|---|---|---|---|---|
| 1-TiC | 63.9 | - | - | 34.5 | 1.4 | - | - | - | - |
| 2-Phospide | 35.5 | 1.0 | 11.3 | 20.1 | - | 14.6 | 2.7 | 13.2 | 1.0 |

The results from the precipitation analysis on LOM images (see Figure 13) show that the area fraction of precipitates averages 0.55 and changes little over the cross section. The area of each precipitate, which was determined by the total precipitate area over the number of precipitates, coincides with the measured SDAS values over the cross section.

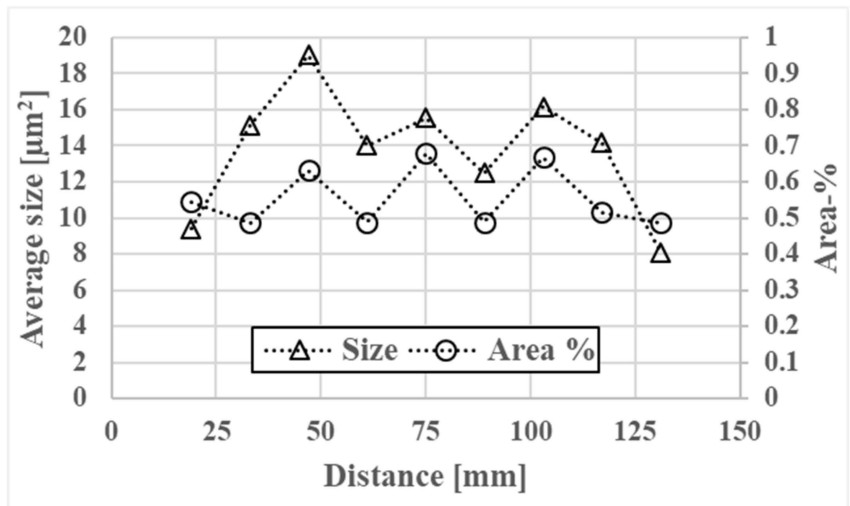

**Figure 13.** Average area and area fraction of precipitates.

Clusters of inclusions, as seen in Figure 14, are often observed together with black particles. These clusters of inclusions are common, close to the surface. However, they are also found infrequently in the whole cross section. Elemental analysis showed that the black particles were $Al_2O_3$-MgO spinel phase and the surrounding orange particles were TiN.

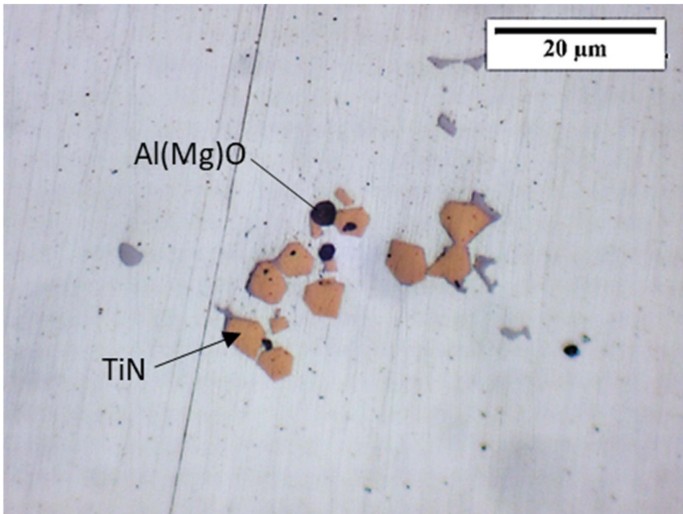

**Figure 14.** Clustered inclusions of TiN in sample 2, taken with LOM in 100× magnification.

### 3.5. Segregation

An example of a line scan from the center of the dendritic area to the center of the interdendritic area is shown in Figure 15. It should be pointed out that precipitates in the interdendritic areas were avoided and not included in the line scan results.

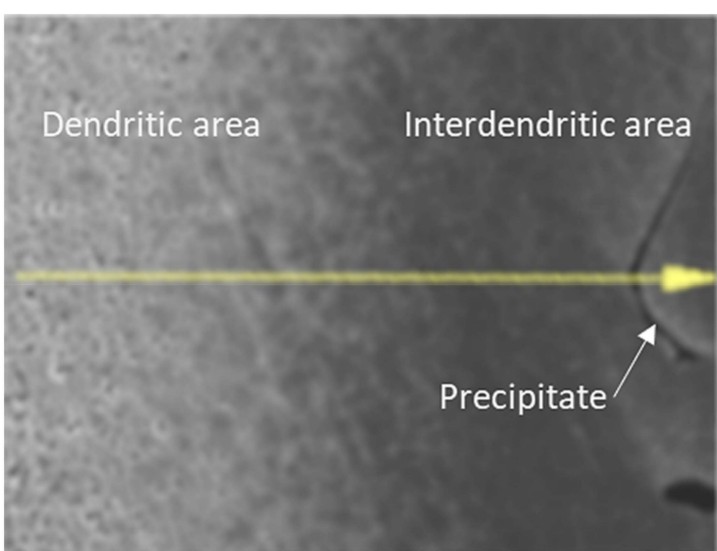

**Figure 15.** Example of an EDX line analysis from dendritic area to the interdendritic area.

The segregation index for each element as a function of distance from billet surface is shown in Figure 16. The element that is most sensitive to segregation is Ti with values between 1.7 to four, followed by Mo, Ni, and Mn also having values above one. However, Cr has values at around one or slightly below. At 30 mm there is a significant drop of the segregation index for Ti and Mo, which happened to be at the same distance as pores appears in the structure. Table 4 shows the difference between the segregation index at 15 mm and 60 mm from the surface, which corresponds to SDAS values of approximately 20 μm and 70 μm, respectively. The segregation indices for Mo, Ni, Mn, and Cr are approximately the same for the two different arm spacings but increases significantly for Ti at larger SDAS values.

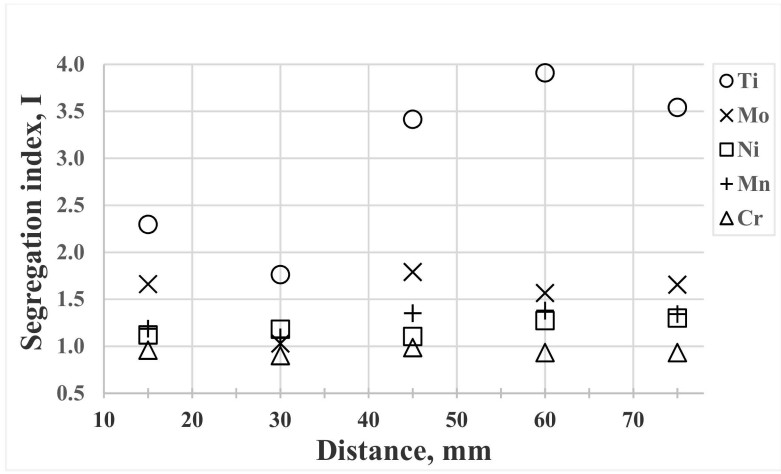

**Figure 16.** The variation of the segregation index measured with EDX plotted against distance from the billet surface to the center.

**Table 4.** Segregation index of alloying elements 15 mm and 60 mm from the billet surface, corresponding to the SDAS values 20 μm and 70 μm.

| Distance | Ti | Mo | Ni | Mn | Cr |
|---|---|---|---|---|---|
| 15 mm | 2.3 | 1.7 | 1.1 | 1.2 | 1.0 |
| 60 mm | 3.9 | 1.6 | 1.3 | 1.4 | 0.9 |

Average line scan measurements of Ti and Mo at 15 mm and 60 mm from the surface are shown in Figures 17 and 18, respectively. The content, in wt %, is plotted in relation to $g_S$, which is defined as the measured distance over the total distance between the center of the dendritic area and the center of the interdendritic area. The images show the difference in segregation found in the columnar zone closer to the surface and the segregation in the equiaxed zone closer to the center. At 15 mm from the surface, the content of Ti goes up from a value of ~1.5 to ~3.3 wt %. However, at 60 mm from the surface the content goes up from a value of ~1.2 to ~5.4 wt %. The distribution of Mo content close to the surface is similar to the content distribution close to the center. Figure 18 shows that for both 15 mm and 60 mm the value goes up from ~0.9 wt % in the dendritic center to ~1.5 wt % in the center of the interdendritic area.

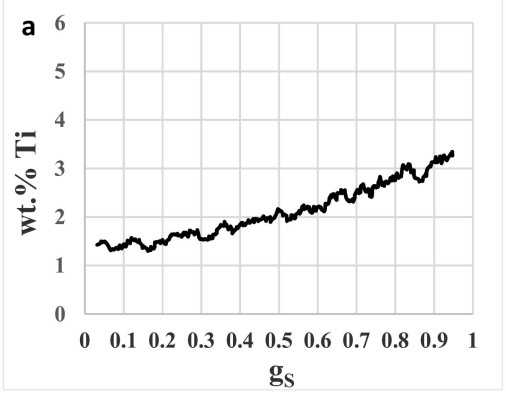
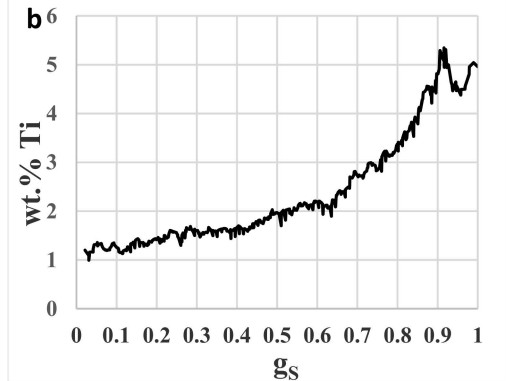

**Figure 17.** Measured Ti content at (**a**) 15 mm and at (**b**) 60 mm from the surface.

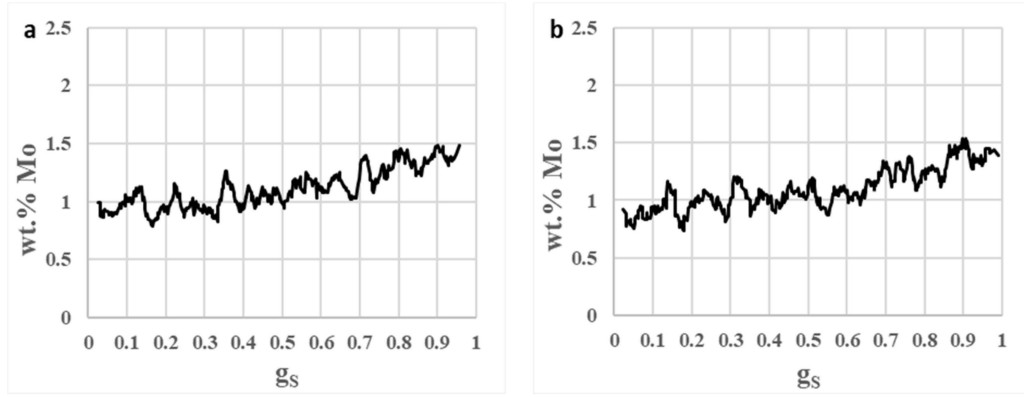

**Figure 18.** Measured Mo content in at (**a**) 15 mm and at (**b**) 60 mm from the surface.

### 3.6. Theoretical Predictions

An equilibrium calculation of the phases versus temperature was calculated using Thermo-Calc TCNI8.1 and the results are shown in Figure 19. The solidification starts at 1387 °C, by the reaction $L \rightarrow \gamma$, and then goes to fully solidified at 1270 °C, which gives the solidification interval between liquidus temperature, $T_L$, and solidus temperature, $T_S$, to be 117 °C. A Ti(CN) phase already exist in the melt at 1550 °C, but increases continuously during solidification from 0.2% up to 0.55%. A laves phase, Fe$_2$Ti, start to precipitate at 960 °C and η-Ni$_3$Ti begins at 750 °C. The Ti(CN) phase composition is shown in the right image in Figure 19, which shows that Ti(CN) precipitate mainly as a nitride above the liquidus. During solidification of $\gamma$, the composition shifts to include more C than N.

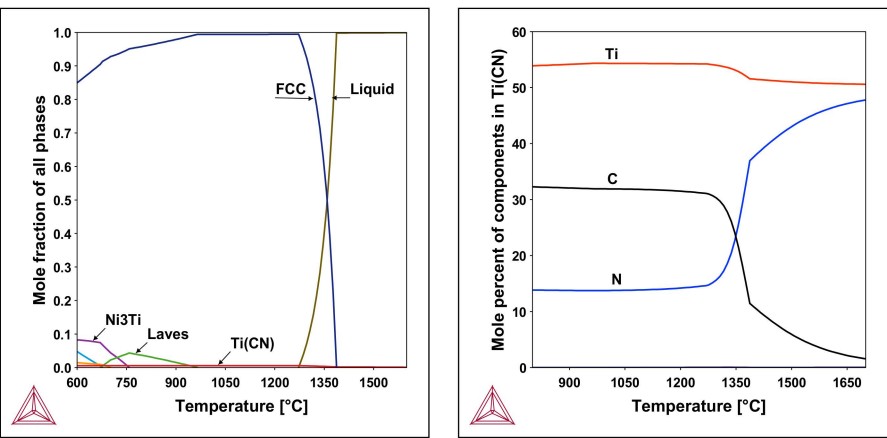

**Figure 19.** Equilibrium phases and composition of the Ti(CN) phase as calculated with Thermo-Calc, database TCNI8.1.

## 4. Discussion

Large pores exist in the center of the billet and the pores are up to 10 mm in width. These are similar porosity values of the same range as found in continuously cast billets of high carbon steels with columnar structure in the center [18]. These carbon steels have a large solidification interval of 40–60 °C [19]. The main reason for the large porosity in the case of A286 alloys is the large solidification interval, which was calculated to be as high as 117 °C, i.e., twice that of high carbon steel grades. Porosity in this range could be troublesome during forging and rolling processes and cause internal cracking and residual porosity.

Precipitates are mainly located in the interdendritic areas. The size of precipitates increases as the structure coarsens and then decreases closer to the center, which is consistent with the measured SDAS values. The equilibrium calculations using Thermo-Calc showed an amount of MC precipitates between

0.5% and 0.7%, which is similar to the amount of precipitates found in the image analysis, 0.5% to 0.7%. The TiC phase was found in the segregated interdendritic areas and in the grain boundaries in the form of flake-like precipitates. The elemental spot analyses showed that these consisted mainly of Ti, C, and around 2 atom% Mo. The morphology indicates a solid phase precipitation.

The Thermo-Calc calculation of components of the MC-phase above the liquidus temperature shows a TiN-phase which corresponds to the cubic orange inclusions in Figure 14. During cooling to room temperature these inclusions should change their composition by replacing N with C according to Figure 19. However, this did not happen, probably due to a fast cooling rate which limits the time for element exchange by diffusion.

The non-metallic spinel phase $Al_2O_3$-MgO were found in many TiN precipitates. It has been reported that spinel can act as inoculant for Ti(CN) [20,21].

The nickel-rich bright grey phase found in segregated areas, see Figure 10, is presumed to be η-$Ni_3$Ti based on the analyzed Ni/Ti ratio and the acicular platelet morphology, which was formed in solid phase. Thermodynamic calculations show that η-$Ni_3$Ti is formed after solidification below 750 °C.

Phosphides were found in the last solidified melt. The composition of these phosphides varies, but the concentrations of P and S stands out. The phosphide precipitates could be a low melting point eutectic of a $M_3$P phase, which has been found both in high manganese austenitic steels and electrolytic iron alloyed with P [13,22]. The presence of regular phosphides in the structure also indicates the possibility of P rich segregate layers forming in grain boundaries, which cause grain boundary decohesion [22]. Since these layers are only a few atomic layers in thickness they would be difficult to detect by using EDX analysis.

Continuous casting of stainless steel blooms that solidify primarily as austenite displays a columnar structure from the surface to the center [23]. The alloy A286 also solidify as primary austenite and it was expected that the structure would be a 100% columnar structure. However, the results in Figure 6 shows that an equiaxed zone exists in the center. During the investigation of the structure it was found that TiN precipitates existed in the whole cross section. However, these particles are not effective inoculants of austenite [20,24]. Particles of spinel were also found and if these particles are not covered by Ti(CN), they act as inoculants for austenite and promote the formation of an equiaxed structure [24].

There are some anomalies at the billet surface region. Figure 5a,b shows that there is a coarse dendrite structure and what looks like an oscillation mark. The most probable reason is the formation of a folding oscillation mark, giving a lower heat transfer and thus a coarser structure.

The secondary dendrite arm spacing, SDAS, is known to affect properties such as the strength and hardness of a material. Moreover, it affects the soaking time needed to reduce microsegregations, as well as the time to dissolve possibly detrimental phases precipitated in the interdendritic regions. The relation between SDAS and cooling rate, $dT/dt$, is usually expressed [25–27] as follows:

$$SDAS = K\left(\frac{dT}{dt}\right)^n \tag{3}$$

where, the value of $K$ is in $K$/s, and n is dimensionless. Usually n has values between −0.33 to −0.5 [25,26]. An investigation of the solidification structures of several Fe-Ni-Cr alloys, ranging from 5 to 20 wt % Cr and 5 to 30 wt % Ni, showed that K varied between 40 to 55 μm and n varied between −0.4 to −0.5 [27]. The constants in Equation (3) are determined empirically in solidification experiments using controlled cooling rates. However, they can be estimated based on measured values of SDAS and a calculated cooling rate. To assess the cooling rate, a simulation of the solidification of the shell was done by assuming one-dimensional heat flow. Thereafter, Fourier's second law was solved numerically using the software Comsol Multiphysics 5.4. The assumption is valid at small shell thicknesses.

The heat transfer was matched so as to keep the surface temperature, $T_{surface}$, at 1000 °C throughout the simulation. Thereby, the cooling rate could be assessed by using the following equation:

$$\frac{dT}{dt} = \frac{T_L - T_S}{t_f} \tag{4}$$

where, $t_f$ is the local solidification time. The parameters used in the simulation are shown in Table 5.

**Table 5.** Parameters used in the solidifying shell simulation, where k is the heat conductivity, ΔH is latent heat, and $c_p$ is the specific heat.

| Casting Speed [m/min] | $T_L$ [°C] | $T_S$ [°C] | $T_{surface}$ [°C] | $k_{solid}/k_{liquid}$ [W/(m*K)] | ΔH [kJ/kg] | $c_{p, solid}/c_{p, liquid}$ [J/(kg*K)] |
|---|---|---|---|---|---|---|
| 2 | 1387 | 1270 | 1000 | * 30/26 | * 260 | * 0.55/0.72 |

* data from a 316 stainless steel [28].

The solid and liquid temperature lines are plotted against the shell thickness in Figure 20a. The width of $t_f$ increases with increasing shell thickness. In Figure 20b the calculated cooling rate is plotted against the measured SDAS values. A linear curve fitting, dotted line in the figure, was applied to evaluate the values for K and n. The obtained values were K = 31.7 µm and n = −0.38.

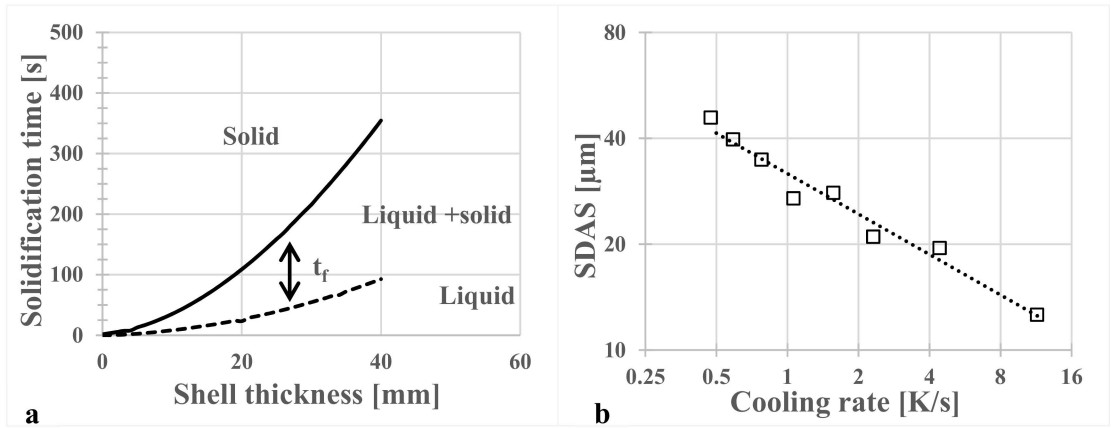

**Figure 20.** (**a**)The calculated solidification time against shell thickness and (**b**) measured secondary dendrite arm spacing (SDAS) against calculated cooling rate.

Porosities starts to appear in the interdendritic areas at 30 mm from the billet surface, as shown in Figure 9. The fraction of porosity, $f_{porosity}$, in the columnar structure is about 0.001, which corresponds to a solid fraction, $f_s$, of 0.999. The liquid feeding in the mushy zone towards the billet surface is almost complete in order to compensate for the solidification shrinkage:

$$\beta = (\rho_s - \rho_L)/\rho_L \tag{5}$$

where, $\beta$ is the solidification shrinkage, which usually has a value of 0.03 for steel, $\rho_s$ is the solid density, and $\rho_L$ is the liquid density. By having a $\beta$ value it is possible to calculate the fraction of liquid, $f_L$, that is necessary to create the measured area fraction of porosity when liquid feeding has stopped as follows:

$$f_L = \frac{f_{porosity}}{\beta} = \frac{0.001}{0.03} = 0.033 \tag{6}$$

The pressure gradient, $dP/dx$, in the mushy zone is the driving force for the interdendritic liquid flow, $v_L$, which is estimated using Darcy's law [29]:

$$v_L = -\frac{K}{\mu \cdot f_L} \cdot \frac{dP}{dx} \tag{7}$$

where, $K$ is permeability (m$^2$) and $\mu$ is dynamic viscosity (kg/m·s).

Heinrich and Poirier [30] have proposed the following expression for the permeability:

$$K = 3.75 \cdot 10^{-4} \cdot f_L^2 \cdot \lambda_1^2, f_s \geq 0.35 \tag{8}$$

where, $\lambda_1$ is the primary dendrite arm spacing [m]. By combining Equations (7) and (8), the pressure gradient in the mushy zone can be expressed as:

$$\frac{dP}{dx} = \frac{v_L \cdot \mu}{3.75 \cdot 10^{-4} \cdot f_L \cdot \lambda_1^2} \tag{9}$$

The liquid flow can be estimated by the following relation [31]:

$$v_L = \beta \cdot \frac{ds}{dt} \tag{10}$$

where, $ds/dt$ is the velocity of the solidification shell obtained from the solid line in Figure 20.

It is well known that a large solidification interval, $T_L$-$T_s$, leads to an extended mushy zone entailing more resistance for liquid feeding. Figure 21a shows a plot of $f_s = 1$ to $f_s = 0.4$ versus the distance from the billet surface at the times 35 s, 110 s, 220 s, and 350 s. The colored lines denote the times at which the shell is completely solidified, $f_s = 1$, at the distances 10 mm, 20 mm, 30 mm, and 40 mm. The feeding distance through the mushy zone, $f_s < 1$, indicated with arrows, increases further in from the billet surface. It is almost four times longer for the 40 mm distance as compared with the 10 mm distance. This increase in feeding distance could explain the start of a porosity formation at 30 mm. Equation (9) was solved numerically to study how the pressure varies in the mushy zone at the distance 30 mm. When the pressure drops below the equilibrium pressure for gas in the liquid, conditions for pore formation exist. The input values of parameters for the calculations are given in Table 6.

Table 6. Input values of parameters to solve Equation (9).

| Shrinkage β | $V_L$ [m/s] | $\lambda_1$ [μm] | Dynamic Viscosity [kg/m·s] |
|---|---|---|---|
| 0.03 | $1.74 \times 10^{-4}$ | 364 | 0.005 |

The results of the calculations are presented in Figure 21b as the pressure plotted verses the distance in the mushy zone and solid fraction. The pressure drops sharply when $f_s > 0.97$ i.e., very close to the position, within 1.5 mm, where the shell has solidified completely. This value correlates well to the calculated value of the fraction of liquid at the start of porosity formation at 30 mm distance from the surface. Thus, conditions exist for gas pore formation. For values of $f_s < 0.97$, there seems to be almost no resistance to a liquid flow, due to the very small pressure drop.

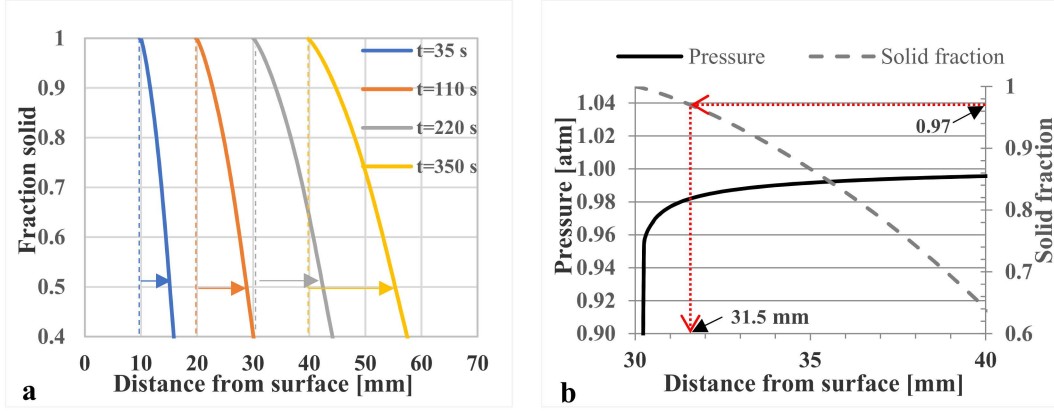

**Figure 21.** (**a**)The fraction solid is plotted against the distance from the billet surface at the times 35 s, 110 s, 220 s, and 350 s. (**b**) Pressure and solid fraction against distance in the mushy zone.

In the equiaxed zone the area% of porosity sharply increases to a value of 0.15%. This can be attributed to the lower permeability and longer liquid feeding distance in an equiaxed zone.

The results in Figure 16 show that the segregation index for Ti increases from approximately 2 to 3.9 at a distance 45 mm from the billet surface. One reason for the increase is the start of the equiaxed zoned at approximately 50 mm from the billet surface. However, the same behavior cannot be seen for the Mo segregation index, which is nearly constant from the surface to the center. At this time it is not possible to determine the reason for the sharp increase of the segregation index of Ti. This high Ti segregation could adversely affect the hot ductility of the billets center area.

The homogenization time prior to hot rolling is one important process parameter to be determined to improve the hot ductility. The measured SDAS values in Figure 6 are used to estimate the homogenization time from the surface to the center of the billet. We considered the relation between diffusion time, *t*, and SDAS, $\lambda_2$, by using Einstein's Brownian motion equation expressing the diffusion time as follows:

$$\lambda_2 = \sqrt{2Dt} \tag{11}$$

where *D*, is the diffusion constant and *t* is the homogenization time, which will give a rough estimate of the time it takes to homogenize the cast product. The difference in diffusion time in the cross section is written as the ratio between the SDAS value at an increasing distance, *x*, from the surface, $\lambda_2(x)$, and the SDAS value at the surface, $\lambda_2(0)$, such as:

$$\frac{t_x}{t_0} = \left(\frac{\lambda_2(x)}{\lambda_2(0)}\right)^2 \tag{12}$$

Figure 22 shows the results from calculations using Equation (12). The results show that it would take approximately 30 times longer to homogenize the structure at 60 mm than at the billet surface.

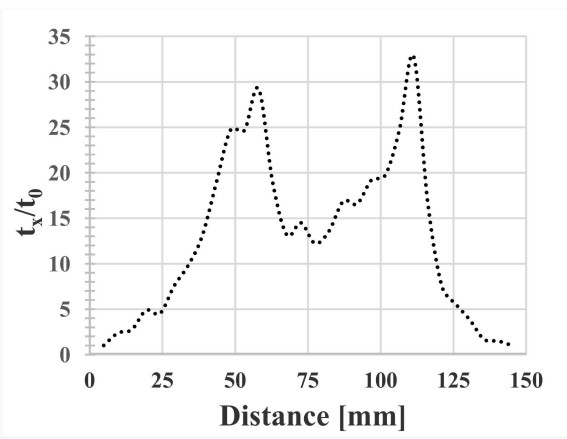

**Figure 22.** Relative homogenization time in the billet cross section.

## 5. Conclusions

The structure and segregation of the A286 alloy in a continuously cast billet, with a dimension of $150 \times 150$ mm, have been investigated. Furthermore, some thermodynamic calculations of stable phases and numerical calculations of the temperature distribution during casting were made. The specific conclusions of the investigation are summarized as follows:

- The billets contain a large central porosity with a width of up to 10 mm.
- Formation of small microporosities in the interdendritic areas start at 30 mm from the billet surface and continue inward to the billet center.
- The investigations with Darcy's law show that the pressure drops significantly when $f_s > 0.97$. However, when $f_s < 0.97$ there is almost no resistance to liquid feeding.
- All samples contain Ti(C,N), TiN, phosphides, and η-phase precipitates and the area fraction of those phases amounts to approximately 0.55% of the whole specimen.
- The presence of heavy microsegregation of Ti to interdendritic areas gives segregation index values of up to 3.9. Probably this will deteriorate the hot ductility properties of the cast billets.
- The secondary dendrite arm spacing, $\lambda_2$ in μm, was measured from the billet surface to the center. A curve fit of the SDAS values versus calculated cooling rates, dT/dt in K/s, gave the relation $\lambda_2 = 31.8 \cdot (dT/dt)^{-0.38}$.
- The center of the billet exhibits an equiaxed structure. The reason for this could be the presence of spinel particles acting as inoculants.

**Author Contributions:** Conceptualization, B.R.; methodology, B.R. and R.F.; validation, B.R. and R.F.; formal analysis, B.R. and R.F.; investigation, R.F.; writing—original draft preparation, R.F.; writing—review and editing, B.R. and N.A.; visualization, R.F.; supervision, B.R. and N.A.; project administration, B.R. and N.A.; and funding acquisition, B.R.

**Funding:** This research was funded by the Swedish Governmental Agency for Innovation Systems, VINNOVA 2016-02017, where the project funding was given as part of the Metalliska material program.

**Acknowledgments:** The authors wish to thank the Hugo Carlsson research foundation for financial support of our study.

**Conflicts of Interest:** The authors declare no conflict of interest.

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
