# Peer review of "Cast Structure in Alloy A286, an Iron-Nickel Based Superalloy"

_metals, doi:10.3390/met9060711_

Reviewer 1 Report

Dear Authors,

thank for this interesting contribution. The article deals with a special iron-nickel based superalloy of which the essential characteristics are being investigated especially in terms of solidification, porosity, precipitates. The work is very broad, especially in relation to the part of the analysis of the results, discussion and conclusions. The introductory part does not have the same precision, where many statements are not related to specific references, nor the part of the method that could be extended by offering greater value to the article. There are some language oversights, but also numerous errors related to the document format. Several suggestions are included in the attached document.

Author Response

Response to Reviewer 1 Comments

Point 1: The work is very broad, especially in relation to the part of the analysis of the results, discussion and conclusions. The introductory part does not have the same precision, where many statements are not related to specific references, nor the part of the method that could be extended by offering greater value to the article. There are some language oversights, but also numerous errors related to the document format. Several suggestions are included in the attached document.

Response 1:

Most of the reviewer’s comments and suggestions have been adapted in the manuscript. Some references have been added in the introduction section.

Reviewer 2 Report

The present paper deals with the cast structure in alloy A-286, an iron-nickel based superalloy.

The reviewer asks some clarifications on the following points:

- In the Abstract, except for the frist two sentences, it is not explained what the authors did in this work. In the sense that, after the first two sentences, the authors wrote directly conclusions of their study. The authors gave numerical data and precise formula, this is not what the abstract should be. The reviewer warmly suggests to the authors to completely rewrite the abstract, i.e. instead of giving numerical data and formula, they can simply write " The effect of central and micro porosity is taken into account. Moreover a mathematical model has been obtained from the cooling rate data.", and so on.

- In the Introduction section the reference numbers are not cited in order of appearance. Moreover, the are too many paragraphs with indentations, and in this way the introduction section looks like it is made with a lot of "copy and paste paragraphs". The reviewer suggest to well organize the appearance of the introduction section.

- In the Introduction section it is mentioned the equation 1, and after that some specific coefficients values are given. In the reviewer opinion, this formula and specific coefficent values are not well placed in this section. Conversely they should be cited and explained better in a theoretical or numerical section.

-The reviewer suggests a general check in the whole manuscript about the use of acronyms. The acronyms should be explained at their first use, i.e. at line 24 (introduction) the acronym "FCC" is not explained, or at line 64 (experimental work) "CC", or at line 76 (experimental work) "LOM" and "SEM/EDX" are not explained.

- The appearance and style of all the equations presented in manuscript is strange. Please update all them, correcting the font size and erasing the cell grid.

- In the "Discussion" section, the caption of the Figure 21(a) is not properly explaining what is plotted. On vertical axis is represented the Fraction Solid, and on the horizontal axis is depicted the Shell Thickness. Therefore why there are four different colored curves in the plot? What the different colors represent?

- In the "Discussion" section, it is depicted the Figure 21(a). In the whole manuscript there is not cited the Figure 21(a), differently it is cited Figure 21(b). If this Figure is not useful erase it, otherwise it should be cited and described in the manuscript. The reviewer suggests a general check of the Figures and Tables in the manuscript.

After these major revisions, the present paper can be re-considered for publication.

Author Response

Response to Reviewer 2 Comments

Point 1: In the Abstract, except for the first two sentences, it is not explained what the authors did in this work. In the sense that, after the first two sentences, the authors wrote directly conclusions of their study. The authors gave numerical data and precise formula, this is not what the abstract should be. The reviewer warmly suggests to the authors to completely rewrite the abstract, i.e. instead of giving numerical data and formula, they can simply write " The effect of central and micro porosity is taken into account. Moreover a mathematical model has been obtained from the cooling rate data.", and so on.

Response 1:

The reviewer’s suggestion was adapted in the manuscript

Point 2: In the Introduction section the reference numbers are not cited in order of appearance. Moreover, the are too many paragraphs with indentations, and in this way the introduction section looks like it is made with a lot of "copy and paste paragraphs". The reviewer suggest to well organize the appearance of the introduction section.

Response 2:

Some indentations removed along with some restructuring.

Point 3: In the Introduction section it is mentioned the equation 1, and after that some specific coefficients values are given. In the reviewer opinion, this formula and specific coefficent values are not well placed in this section. Conversely, they should be cited and explained better in a theoretical or numerical section.

Response 3:

The section was replaced in the discussion part together with other theoretical investigations.

Point 4: -The reviewer suggests a general check in the whole manuscript about the use of acronyms. The acronyms should be explained at their first use, i.e. at line 24 (introduction) the acronym "FCC" is not explained, or at line 64 (experimental work) "CC", or at line 76 (experimental work) "LOM" and "SEM/EDX" are not explained.

Response 4:

The reviewer’s suggestion was adapted in the manuscript.

Point 5: The appearance and style of all the equations presented in manuscript is strange. Please update all them, correcting the font size and erasing the cell grid.

Response 5:

The reviewer’s suggestion was adapted in the manuscript.

Point 6: In the "Discussion" section, the caption of the Figure 21(a) is not properly explaining what is plotted. On vertical axis is represented the Fraction Solid, and on the horizontal axis is depicted the Shell Thickness. Therefore why there are four different colored curves in the plot? What the different colors represent?

Response 6:

The reviewer’s suggestion was adapted in the manuscript. Figure 21a was wrongly cited as Figure 22a in text above the figure. Section with explanation moved to be more related with the figure.

Point 7: In the "Discussion" section, it is depicted the Figure 21(a). In the whole manuscript there is not cited the Figure 21(a), differently it is cited Figure 21(b). If this Figure is not useful erase it, otherwise it should be cited and described in the manuscript. The reviewer suggests a general check of the Figures and Tables in the manuscript.

Response 7:

The reviewer’s suggestion was adapted in the manuscript. Figure 21a was wrongly cited as Figure 22a in text above the figure.

Reviewer 3 Report

Authors present quite large investigation of Ni-based superalloy after casting. The study is clear but not so attractive to give new information and ideas for the community. Text is too long and fragmented. i suggest to rewright text to make it more readable and clear for readers. There are missing parts (references and explanations). Authors give 22 figures, 12 equations and 6 Tables. it is too much for such exploratory study. I suggest authors to work more careful on the text and visual as well as number data. In particular:

Line 61. Should be "Matherials and Methods".

All eqations should be formated and not inserted as tables.

Line 17. "phosphides". What kind of phases? Their composition, structure etc.?

Line 56. "The purpose of this paper..." should be "In the current study,..."

Table 1 contains too less information, it is probably better to give all numbers as text.

Abbreviatoins such as LOM, CC, ESR, PDAS, should be systematically given at the first place of appearence.

Fig. 11 and Table 2. Authors give also carbon maps as well as carbon content in the table for area 3. I am not sure that carbon can be realistically qualified and quantified using EDX without special techniques. I suggest to remove carbon from the discussion or do some special efforts to perfrom more precise analysis. What is the origin of carbon? Did authors performed special manipulations to remove parasitic carbon?

Line 129 and below. "Sample 1 and sample 6" as well as "Samples 2-4" (line 137) there is no exact information about "samples". Authors should give complete information about the nature of all "samples". All diffrences in their preparation, composition and properties. As well as it is not so easy to read text with "samples x" without any idea about them. Please use specifications such as for example "Ni-reach sample 1" or "heat treated sample 2" etc. Or it is just different parts of the same plate according to Fig. 1? If so authors have only one sample and 11 spots on it...

Fig. 18. Should be wt.% but not mass%, it is more common.

All references are missing in the text which makes difficult to review.

Thermo-Calc modelling should be accompained with a reference to TCNI8.1 database. I also suggest to give a short sentance about the database. Is it suitable for current superalloy?

Author Response

Response to Reviewer 3 Comments

Authors present quite large investigation of Ni-based superalloy after casting. The study is clear but not so attractive to give new information and ideas for the community. Text is too long and fragmented. i suggest to rewright text to make it more readable and clear for readers. There are missing parts (references and explanations). Authors give 22 figures, 12 equations and 6 Tables. it is too much for such exploratory study. I suggest authors to work more careful on the text and visual as well as number data. In particular:

Response:

In response to the attractiveness of this study: This A286 alloy is continuously cast, the common method for manufacturing is by ingot casting and then ESR remelting. By changing the process route there is a large increase in energy effectiveness.

Point 1: Line 61. Should be "Matherials and Methods".

Response 1:

The reviewer’s suggestion was adapted in the manuscript.

Point 2: All eqations should be formated and not inserted as tables.

Response 2:

The reviewer’s suggestion was adapted in the manuscript.

Point 3: Line 17. "phosphides". What kind of phases? Their composition, structure etc.?

Response 3:

Although the composition has been analysed, it could not be decided which type of phase these phosphides belongs, therefore the general description.

Point 4: Line 56. "The purpose of this paper..." should be "In the current study,..."

Response 4:

The reviewer’s suggestion was adapted in the manuscript.

Point 5: Table 1 contains too less information, it is probably better to give all numbers as text.

Response 5:

It is common to give the materials composition in the form of a table for readability.

Point 6: Abbreviatoins such as LOM, CC, ESR, PDAS, should be systematically given at the first place of appearence.

Response 6:

The reviewer’s suggestion was adapted in the manuscript.

Point 7: Fig. 11 and Table 2. Authors give also carbon maps as well as carbon content in the table for area 3. I am not sure that carbon can be realistically qualified and quantified using EDX without special techniques. I suggest to remove carbon from the discussion or do some special efforts to perfrom more precise analysis. What is the origin of carbon? Did authors performed special manipulations to remove parasitic carbon?

Response 7:

Some clarifications are now made regarding the carbon content.

Point 8: Line 129 and below. "Sample 1 and sample 6" as well as "Samples 2-4" (line 137) there is no exact information about "samples". Authors should give complete information about the nature of all "samples". All diffrences in their preparation, composition and properties. As well as it is not so easy to read text with "samples x" without any idea about them. Please use specifications such as for example "Ni-reach sample 1" or "heat treated sample 2" etc. Or it is just different parts of the same plate according to Fig. 1? If so authors have only one sample and 11 spots on it...

Response 8:

The sample designation is clarified by the following sentence on line 70:

“To produce micro samples, a slice was cut from the plate and then divided into cross-sectional samples designated 1-11, where sample 6 included the billet centre.” In this case the samples are defined as micro samples taken from different locations on the cut-out shown in Figure 1.

Point 9: Fig. 18. Should be wt.% but not mass%, it is more common.

Response 9:

The reviewer’s suggestion was adapted in the manuscript.

Point 10: All references are missing in the text which makes difficult to review.

Response 10:

It is unclear what the reviewer refers to, all references are indeed visible in the text, e.g. “..uniform formation of strengthening precipitates [1].”.

Point 11: Thermo-Calc modelling should be accompained with a reference to TCNI8.1 database. I also suggest to give a short sentance about the database. Is it suitable for current superalloy?

Response 11:

The reviewer’s suggestion was adapted in the manuscript.

Round  2

Reviewer 2 Report

The present paper deals with the cast structure in alloy A-286, an iron-nickel based superalloy.

The reviewer asks some clarifications on the following points:

-The reviewer suggests a general check in the whole manuscript about the use of symbols. For example, in Equation 2 the solidification shrinkage depends on two parameters, the reviewer didn't find the definition of the two parameters ( maybe densities ). Please explain if the reviewer didn't see them, or update the manuscript.

- The captions of the Figure 2 and Figure 21(a) are not properly placed, too close to the pictures, please update them.

- In Figure 21(a) it is not clear what is plotted. On vertical axis is represented the Fraction Solid, and on the horizontal axis is depicted the Shell Thickness. Therefore why there are four different colored curves in the plot? What the different colors represent? Just seeing at the picture, the solid fraction can be two different values for a shell thickness of 40 mm? It is 1 for the yellow curve and 0.6 for the grey curve. It is an unique shell structure, or they are different shell structures with different thicknesses? This figure is not clear.

After these minor revisions, the present paper can be re-considered for publication.

Author Response

Response to Reviewer 2 Comments

Point 1: The reviewer suggests a general check in the whole manuscript about the use of symbols. For example, in Equation 2 the solidification shrinkage depends on two parameters, the reviewer didn't find the definition of the two parameters (maybe densities). Please explain if the reviewer didn't see them, or update the manuscript.

Response 1:

The definition of the densities has been added to the manuscript.

Point 2: The captions of the Figure 2 and Figure 21(a) are not properly placed, too close to the pictures, please update them.

Response 2:

The caption for Figure 2 and Figure 21(a) has been updated.

Point 3: In Figure 21(a) it is not clear what is plotted. On vertical axis is represented the Fraction Solid, and on the horizontal axis is depicted the Shell Thickness. Therefore why there are four different colored curves in the plot? What the different colors represent? Just seeing at the picture, the solid fraction can be two different values for a shell thickness of 40 mm? It is 1 for the yellow curve and 0.6 for the grey curve. It is an unique shell structure, or they are different shell structures with different thicknesses? This figure is not clear.

Response 3:

The authors thank the reviewer for helping  to make this clearer to the reader.

Clarification has been made in the text as:

“Figure 21a shows a plot of fs=1 to fs=0.4 versus the distance from the billet surface at the times 35 s, 110 s, 220 s and 350 s. The colored lines denote the times at which the shell is completely solidified, fs=1, at the distances 10 mm, 20 mm, 30 mm and 40 mm. The feeding distance trough the mushy zone, fs<1, indicated with arrows , increases further in from the billet surface.”

Figure is updated with legend and caption is updated.

Reviewer 3 Report

Authors significantly improved their text. It might be published in the current form.

Author Response

Response to Reviewer 3 Comments

Point 1: Authors significantly improved their text. It might be published in the current form.

Response 1: The reviewer’s comment is kindly received.